mental health care in West Africa; prayer camps; global mental health (GMH); movement for global mental health; critique of the movement for global mental health (MGMH); mental health and human rights; social psychiatry in West Africa

**Corresponding author:**
Michael Huppertz;
Email: mihup@web.de

# A response to criticism of the global mental health movement. How polarization can be overcome in theory and in west African social psychiatric practice

Michael Huppertz[1,2]

[1]Psychiatrist and Psychotherapist, Sociologist, Kreuz 10A, D 85625 Glonn, Germany and [2]Executive Board, Mindful Change Foundation, Darmstadt, Germany

## Abstract

Since the turn of the 21st century, we have seen the development of an international movement that works in various ways to ensure that everyone in the world has access to adequate mental health care. There is indeed a great need for action, especially in countries with weak and underfunded health systems. The Movement for Global Mental Health (MGMH) is supported by strong organizations such as the WHO, academic institutions and NGOs. As this movement has gained momentum, however, it has been accompanied by fierce criticism, in particular from scholars of the humanities and social science, who see the global expansion of psychiatry as a medical discipline as a form of power-grabbing, neocolonialism and capitalist expansion. They also consider psychiatry to be a biologistic discipline, the justification of which they question, in continuation of a long anti-psychiatric tradition. This criticism prompted several adaptations of the MGMH and various efforts towards integration, but these have not been widely accepted by the critics. The following text primarily summarizes, classifies and critically engages with the basic arguments of the aforementioned critique. Theoretical misconceptions regarding the practice of psychiatry are clarified. Subsequently a specific project in Côte d'Ivoire is presented that demonstrates how contextual psychiatry can proceed and how unnecessary dichotomies and polarizations can be overcome in the interests of the persons concerned.

## Impact statements

The path to Global Mental Health is neither a lane on a neo-colonialist highway nor does it run from theory to practice. The Movement for Global Mental Health can respond constructively to its critics by understanding psychiatry as a lively network of people, things, practices and ideas. It consists of, among other things

- the experiences and concerns of patients and families,
- the interests and capacities of communities and societies,
- experiential knowledge, traditional and local interpretations, sciences and humanities,
- non-human structures (infrastructure, climate, etc.),
- a non-reductive bio-psycho-social model of illness,
- existing traditional and medical care structures and practices.

This view suggests that different individual and cultural perspectives are unavoidable and indispensable and can lead to controversial debates and productive cooperation. I do not wish to call into question the constructive achievements and contributions that critics of the MGMH have achieved. I am interested in addressing the unusually harsh criticism of the MGMH. I would like to clarify some of the fundamental misunderstandings that underlie this polarizing campaign in order to soften the blow. This also seems so important to me because I see so much common ground in the fundamental concerns. Cooperations should bring together the people involved in this topic in order to tackle one of the most urgent tasks in international health care and to combat a structural human rights catastrophe.

## Introduction

In the 1980s and 1990s, the author was involved in the de-hospitalization of long-term psychiatric patients and the development of outpatient care. These were major concerns of the social psychiatric movement in Europe. He felt particularly connected to the Italian psychiatric reform through visits and internships. The movements in these countries pursued a "principle of greatest

need": commitment to the sickest, poorest and most isolated. But the interest and commitment stopped at Europe's borders. In 2015, he learned about the plight of mentally ill people in Indonesia and West Africa from journalists' reports. He traveled to Bali and the Ivory Coast. In Ivory Coast, he began working with Prof. Koua, who wanted and still wants to develop community-based psychiatry in the country. In 2018, the author and his colleagues founded a non-profit foundation in Darmstadt, Germany, the Mindful Change Foundation, to support social psychiatric model projects in countries without psychiatric care for the vast majority of the population. The undesirable developments in European psychiatry should be avoided and modern, patient-and community-oriented psychiatry should be developed that meet local conditions.

The author is a psychiatrist and sociologist and holds a doctorate in philosophy. He was particularly specialized in phenomenological psychiatry and was always involved in the discussion about social and anti-psychiatry. When he became aware of the Movement for Global Mental Health, he was astonished that an essentially anti-psychiatric position dominated the discussion on this topic within the social and cultural sciences. These attacks against the MGMH led the Movement to make some corrections, but it seems that the fundamental questions of the debate have not been sufficiently addressed and resolved. This weakens the global commitment to people with mental illness and epilepsy. Although not the intention of the critics of the MGMH, their criticism entails the risk of public intellectual and political leaders preferring to focus on less controversial issues than Global Mental Health and justifying this neglect by voicing the arguments of the MGMH's opponents.

The aim of this text is to bridge the gap between the two movements through a modern understanding of psychiatry. It should overcome the biologism in psychiatry as well as the some-times individualistic, sometimes sociologistic–culturalistic concepts of the MGMH's critics. This is possible if necessary practical development goes hand in hand with modern philosophical concepts (pragmatism, realism, actor-network-theory) that understand human thought and action as part of complex networks in which infrastructural and social conditions play a significant role and concepts and individual particular interests do not dominate.

Of course, our concepts and categories form and narrow our view of the world. In doing so they also create differentiations, connections, terms and advance cognition. They are certainly a form of exercising power through demarcation and exclusion. But this function is often overestimated and to generalized in today's post-structuralist cultural studies. Following Foucault, it has become common to understand every form of discourse mainly as a form of power. I think this is a hindrance to our topic because the resulting assumptions are too crude and do not do justice to the realities on the ground. Concepts direct our interest and differentiate our perception. Linguistic and theoretical concepts also have a pragmatic and heuristic meaning, which only works if they fit the specifications of reality. In this sense, I follow the realistic tendency of current philosophy. The important thing to remember is that reality does not tolerate every way of dealing with it. Our practice, our networks and our perceptions are always more than our concepts. That is why they can change constantly. Even power structures cannot prevent that. This also applies to the medical model and psychiatry. As the practice and institutions of psychiatry have changed considerably in recent decades, so too has its discourse. And if this happens too slowly and if it does not adapt and expand sufficiently with its global expansion, then we must ensure that this happens.

## The movement for global mental health

Since the beginning of the 1990s, the international health movement has developed a particular commitment to mental health. This led to the WHO issuing a detailed programmatic statement in 2001 and a solution-oriented review of findings by a group of researchers who published an important reference text in The Lancet in 2007 (Patel et al. 2007, Patel 2012). The extensive network of parties involved includes research institutions, nongovernmental organizations, individuals and international organizations such as the WHO. Their core beliefs provide a common paradigm, so it is appropriate to speak of a 'Movement for Global Mental Health (MGMH)' (Patel 2012; Patel et al. 2018; Kirmayer, Pedersen 2014). A separate organization of the same name was also founded, uniting numerous individuals and organizations (https://www.globalmentalhealth.org). The findings and demands of the MGMH found their way into WHO programmes, particularly the Mental Health Gap Action Programme (`mhGAP´, WHO 2008). The MGMH follows a universalistic orientation and pursues the following essential goals (see Table 1):

One key criticism of the MGMH is that it obscures the view of the social contexts of both the emergence and interpretation of mental health problems and disorders and that it neglects or even obliterates local traditions (Summerfield 2012, 2013; Mills and Fernando 2014; Mills 2015, 2018; Cosgrove et al. 2019; Bracken et al. 2021). At one extreme, there is talk of a structural identity of psychiatry and colonialism (Mills 2018). This critique has been echoed by the MGMH for several years, emphasizing the importance of local contexts (Patel et al. 2018). Many authors are now calling for a shift away from a primarily biomedical understanding of psychiatry towards more complex research and practice that take local interpretations as well as community-based and grassroots approaches seriously (Campbell and Burgess 2012; Kirmayer, Pedersen 2014; Cooper 2016; Chase et al. 2018; Patel et al. 2018; Pūras 2017; Gone, Kirmayer 2020; Ojagbemi, Gureje 2020; Sugiura et al. 2020; Gómez-Carrilon et al. 2020; Koua 2022). However, this development does little to impress the critics of the MGMH (Cohen et al. 2018; Bracken et al. 2021).

## The criticism of the movement for global mental health

The author has no doubt about the terrible consequences of psychiatric concepts and practices in the past and present, from the `total institutions´ that still exist in many countries today with terrible living conditions, maltreatment and unjustified and unjustified an and excessive privations of liberty. The history and the

**Table 1.** Concerns of the Movement for Global Mental Health (MGMH)

| Concerns of the Movement für Global Mental Health (MGMH) |
| --- |
| • the commitment to the human rights of mentally ill people |
| • The scientificity of psychiatry and the understanding thereof as a medical discipline |
| • The destigmatization of affected persons and their families |
| • The updating of national laws and development of national mental health programmes |
| • The development of decentralized care as an integrated part of general health care |
| • The priority of outpatient over inpatient work |
| • The establishment of inpatient treatment centres/spots/facilities in general hospitals |

**Table 2.** Criticism of the Movement for Global Mental Health (MGMH)

| Criticism of the Movement for Global Mental Health |
| --- |
| The MGMH<br>• represents a biomedical model that is not applicable to mental distress,<br>• adopts post-colonialist and capitalist interests and perspectives,<br>• individualizes suffering and ignores the social and material contexts,<br>• controls and destroys explicitly or implicitly deviant behavior and the subjectivity of the person concerned,<br>• asserts superiority over indigenous interpretations and indigenous ways of dealing with mental distress. Local perspectives are not taken seriously and are marginalized. |

present of psychiatry are questionably motivating for a critique of psychiatry. Even the author will never forget the conditions in the old psychiatric hospitals in the 1980s. He only saw something worse in the Prayer Camps in West Africa. However, he draws different conclusions from these experiences because he considers the central theses of the critics, including their metatheoretical assumptions (see table 2), to be outdated.

In my opinion, the controversy is exacerbated by the fact that there is hardly any international research on the situation of mentally and epileptically ill people and the various approaches to care. This is regrettable in any case. If it existed, it would considerably defuse the present controversy. The lack of research is difficult to understand and accept, as it affects around 4–5% of the population (at any time), assuming 2–3% for severe mental illnesses and around 2% for epilepsy, which is only a rough estimate. The percentages are much higher if post-traumatic stress disorder, addiction and anxiety disorders, autism, etc. are included. Mentioning epilepsy in the same breath as mental illness may seem strange to medically trained people, but it is common in many countries in West and Central Africa because the population makes little or no distinction between epileptic and mental illness. Both show an incomprehensible, strange and sometimes frightening behavior that is obviously not based on free decisions. Both illnesses lead to isolation and stigmatization in a similar way and are rarely treated. Children with epilepsy are not allowed to attend school or sit at the table with their relatives. Epileptic disorders are more common in these countries than in Europe, mainly due to the lack of obstetric care.

### "Western psychiatry" and the "global south"

Of course the institution of science is historically rooted in "western" culture through its practices, language, metaphors and interests. However, critics confuse origination with validity. The origin of views and theories says nothing about their validity. And the global cultural landscape has changed. The criticized psychiatry has long since spread to countries that cannot be counted among the two political-economic constructs. They do not fit the description of the global presence of psychiatry. It is much more chaotic. Do China, Saudi Arabia, Chile or Argentina belong to the global South? Scientifically oriented psychiatry is just as widespread in countries such as China as it is in South Africa, which in turn has little in common with Burkina Faso. In Indonesia, the situation of mentally ill people in rural areas is no better than in Burkina Faso, even though ambitious international psychiatric congresses are held there. The polarization also ignores the huge cultural and economic differences within the individual countries. Abidjan in Côte d'Ivoire (now with over five million inhabitants) is more like London than any Ivorian village. In Abidjan you have a chance of

psychiatric treatment, but only if you have some money. Almost all Ivorian psychiatrists work in this city, the country's only metropolis. Maybe The Human Development Index (HDI see https://www.laenderdaten.de/indizes/hdi.aspx) is much more differentiated. Does not it make more sense to think more regionally, in terms of economic differences snd simultaneities of different living environments?

### The spread of psychiatry in poor countries (LIC, LMIC) pursues capitalist interests, especially those of the pharmaceutical industry

The accusation that capitalist interests are also being pursued with the idealistic neo-colonialist import is inadequate in relation to the current reality in many countries and the problems faced by the MGMH. Perhaps it will be justified at some point when poverty in the LIC ("Low-Income Countries" according to the common, also very rough, classification of the World Bank) and LMIC ("Lower Middle-Income Countries") is largely eradicated. At present, all West African psychiatric staff are very happy to receive only the oldest and cheapest drugs: mainly chlorpromazine and haloperidol as neuroleptics, phenobarbital and carbamazepine as antiepileptic drugs and amitriptyline for severe depression. Lithium etc. should not be considered (also because there are no possibilities for laboratory controls). Even risperidone, which is classified as an essential drug by the WHO, is too expensive and is therefore reserved for patients with severe side effects, if at all. People have to pay for the medication if it is not donated by NGOs, which we do for those most in need. You have to bear in mind that many people in rural areas survive mainly through subsistence farming. Pharmaceutical companies have not yet found their way into the rural regions of West Africa, where the vast majority of people live, and they know why.

### Psychiatry is wrongly regarded as a discipline of medicine. Psychiatry is biologistic and unsuccessful. It is a pseudoscience and has no scientific basis

A central concern of the MGMM is integration into primary health care. A central criticism, however, claims that psychiatry is a medical discipline in name only. Mental health problems cannot be addressed within the framework of the medical model. Critics of the MGMH assume that the medical model espoused by the MGMH is one major reason for neglecting the subjectivity of patients and different cultural ways of life. Therefore, the movement lacks respect for indigenous traditions in countries without or almost without psychiatric care (Summerfield 2012, 2013; Mills, Fernando 2014; Mills 2015, 2018; Cosgrove et al. 2019; Bracken et al. 2021). In so assuming, they focus their criticism on a strictly biomedical model of disease, according to which only biological and physiological causes are relevant to a disease event. However, this model is but one disease model among many (Franke 2012; Nettleton 2021). Critics overlook the fact that environmental influences of various kinds are also generally recognized as medical causes – in toxicology, infectiology, occupational medicine, psychosomatics, and significant areas in psychotherapy and psychiatry (Braveman et al. 2011). Psychiatric staff tend to use a "socio-environmental model of medicine" (Engel 1977, Nettleton 2021). Without such a broader understanding, these professionals would be unable to act. Illness means a threatening, acute or chronic functional loss. This loss must be severe enough to impact everyday life either now or later, to cause suffering or to shorten life. The

causes may lie in an individual but also in the natural and social environment. From a sociological perspective, "disease" and "health" are not natural categories. Medicine is constantly adapting its individual definitions of illness to developments in research, to diagnostic techniques and to social and individual expectations.

From phenomenological psychiatry to the critique of the MGMH, there is a long tradition of emphasizing the importance of the first-person perspective of those affected by social and personal difficulties. But there is no reason to play them off against the third person perspective of medicine. The subjectivity of the patient is also indispensable in medical discourse. Radiological findings, for example, often reveal massive abnormalities but lead neither to a diagnosis nor to a therapy if there are no matching symptoms or adverse consequences and, conversely, someone will be termed ill if he suffers from chronic pain, paralysis or fatigue that cannot be explained by any existing diagnosis or technical finding. Psychopathology in particular depends on what the patient communicates. How else can one distinguish depression from grief or paranoid symptomatology from social anxiety? The historically significant influences of psychoanalysis, phenomenological and existential psychiatry are suppressed by the critics of MGMH. This essential complementarity becomes even clearer where intercultural understanding is necessary and when the person concerned may not be sufficiently well understood due to a lack of shared cultural patterns of interpretation.

The critics of the MGMH paint a distorted picture of psychiatry that is far removed from reality. Psychiatric work with people suffering from severe or chronic mental illnesses in particular is predominantly about coping with the psychosocial aspects such as managing everyday life, going shopping, the desire for a loving relationship, a satisfying job or having a good perspective on life. Sometimes it is about psychotherapy, sometimes it is about medication, sometimes about recovery.

Why not apply the concept of disease to the processes of perception, thought, feeling, communication and behaviour? Critics of the MGMH have no problem with the idea of diagnosis and therapy concerning physical illness. Non-psychiatric diagnoses are even understood in a naïve realist way (Summerfield 2012, 2013; Fernando 2018; Gomory and Dunleavy 2018; Cohen et al. 2018). The reduction of the medical model to a purely biological basis means a return to the long-outdated separation of body and soul. The known numerous interactions and dependencies between somatic and mental processes are ignored. And can we really begin to imagine that mental, psychological and interactive processes and the brain involved in them (Fuchs 2018) can escape all disturbances, dysfunctions? Of all things, can such a complex system as the human psyche with its sensory, cognitive, emotional, social and biological interconnections be exempt from becoming mixed up or from suffering acute or permanent disorders? How could such an assumption be justified?

Of course, the boundaries are blurred when it comes to mental abnormalities and overdiagnosis is a real danger. The therapeutization of society is unquestionably a problem that has been rightly addressed for decades (Rieff 2006 [1966]; Summerfield 2012; Cabanas, Illouz 2019). The fact that the global mental health movement focusses on severe mental illness (SMI; WHO 2008, Mari et al. 2009, Eaton et al. 2014) is often overlooked by critics of the movement. They criticize the inflation of psychiatric diagnoses and therapeutic services without realizing that a terminological problem of their own making clouds their vision. To avoid referring to diagnoses of illness, they speak of "distress" (Summerfield 2012, 2013; Mills, Fernando 2014; Mills 2015; Cosgrove et al. 2019) or "deeply

troubling behaviors and mood states" (Cohen et al. 2018, p. 192). This blurs the lines between the different care settings for mild or severe mental illness and epilepsy. In general, it is important not to lower the threshold for psychiatric diagnoses, but to raise it to conserve resources for those really in need. The diagnosis of a mental illness is always also a social agreement.

Basic training for health center staff in Ivory Coast is to show how to distinguish an epileptic seizure from an acute psychotic symptom. This distinction enables the use of antiepileptic drugs or a neuroleptic if necessary. Prophylaxis looks very different. Thereby a sufficiently correct medical understanding makes it possible to pull a child in an epileptic seizure out of the fire, which usually does not happen because people believe that the spirits the child is possessed by will jump over to the helper. This is why burns are half the diagnosis.

Where deviant behavior is very much considered socially undesirable and threatening, although not simultaneously labeled intentional and criminal, there is a risk of the "culprits" meeting with excessive violence and exclusion. Moreover, the critics do not take into account that overcoming the moral or exclusively spiritual judgment of deviant behavior would be of great benefit for mentally ill and those with epilepsy in many countries in Africa and Asia. Critics speak as little of the positive practical effects of the medical model as they do of the rabid marginalization and widespread – and oftentimes brutal – traditional practices that critics of the MGMH generally gloss over or treat euphemistically (Mills 2018; Cosgrove et al. 2019; Weinmann 2019; Gómez-Carrillo et al. 2020; Bracken et al. 2021). It is hard to imagine a worse exclusion and a worse stigma than being tied outdoors to a tree, to have the lower legs fixed in tree trunks or being confined to a small dark hut for years and without any real human contact.

Within the post-structuralist discourse-analytical tradition, particularly Michel Foucault's representation of the history of psychiatry (1988 [1961]), it is difficult to take it seriously that it was not only the internment of mentally ill people in asylums for outsiders of all kinds in 17th century Europe and the emergence of European psychiatry around 1800 that led to dehumanizing treatment and separation of mentally ill people. Obviously, this separation does not require the categorical distinction between reason and madness within the framework of scientific rationalism. Of course, the lifeworlds and discourses in the countless cultures and subcultures of this planet differ in many ways. But similar fears and dreams are also woven into this tapestry of cultures and times.

However, it is also important that the medical interpretation of deviant behavior be associated with reasonable prospects of humane treatment, recovery or cure. Otherwise, there is a risk of a novel form of stigmatization through the medical discourse (Kvaale et al. 2013). An important question that remains in this context is whether psychiatry has any success at all, especially the most criticized pharmacotherapy. I do not believe that anyone who is halfway familiar with psychiatric practice would seriously claim that all pharmacological intervention is pointless. Every practitioner is aware of their effectiveness in the treatment of acute psychotic symptoms and anxiety states as well as in the prevention of phases of manic-depressive illness. To pick out one important concern there are sufficient high-quality meta-analyses for the benefit of neuroleptics in schizophrenic disorders (Leucht et al. 2013, Tiihonen et al. 2017, Haddad and Correll 2018). Of course, there can be meaningful discussions about dosages, the advantages and disadvantages of individual medications, side effects and, above all, the duration of use. Many mistakes have been and continue to be made here. It also took a long time for more tolerable drugs to be

**Table 3.** Arguments in favor of the medical model in psychiatry

| Arguments in favor of the medical model in psychiatry |
|---|
| • Orientation for patients and their relatives (care, rights, protection, living arrangements, dealing with each other, perspectives, commonality with other affected persons)<br>• Communication between psychiatric professionals (about the patient, exchange of experience, evaluation)<br>• Communication among patients, relatives and psychiatric staff<br>• Communication in interdisciplinary and international research<br>• Integration into general medical care (clarification and treatment of organic medical illnesses with psychological symptoms, psychomatics)<br>• De-moralization of severe mental illness<br>• If therapeutically successful: destigmatization |

developed, such as those available in rich countries today. Research is continuing and it is to be hoped that even better drugs will be developed. These few sentences concern only the most controversial point. However, psychiatric work cannot be reduced to psychopharmacotherapy. After all, all these discussions would have to accept a medical discourse at least temporarily (psychopathology, differential diagnosis, course, comorbidities, side effects, comparison groups, etc.).

Summarized the medical model is a pragmatic concept with many advantages (see table 3):

Defending the utility of the medical model therefore in no way implies that other cultural interpretations and practices are of no interest or importance, not as an ethnological object, but as a partner in dialog and cooperation. But they should not be excluded from critical questions either. Traditional patterns of interpretation and practices can be just as ineffective or harmful, epistemically incorrect and ethically dubious as scientifically oriented ones. Traditional measures can certainly also be helpful, but I vigorously deny – also from my own experience – that this is true in all cases of severe mental illness, and that is what matters. I do not want to support the idea of competition between two essential forms of cultural approaches. The healers and the leaders of the prayer camps also seek help from health centers on their own initiative when they do not know what to do. They also find it difficult to chain people to trees outdoors for years – apart from the fact that it is usually really terrible for the patients and their relatives who ask for help.

### Psychiatry individualizes social suffering. How autonomous are psychological and psychopathological processes?

The critics of the MGMH see individual psychological processes primarily as a resonance chamber for social grievances. The very personal fate, complex inner structures, and the potential inability to handle existential challenges such as questions about one's own identity, attachment and relationship problems, feelings of guilt, or fear of death are superseded by unquestionably weighty issues such as poverty, flight and war (Summerfield 2012; Mills 2015; Cosgrove et al. 2019; Weinmann 2019). Of course, social factors play an important role in every individual fate and to an even greater extent in the fate of groups of people. It is commendable and necessary to emphasize this again and again. The warning against an exaggerated individual attribution of responsibility and risks is no less justified (Beck 1992; Cabanas, Illouz 2019).

However, the appropriation of psychological processes by social processes is dangerous and unrealistic. It de-individualizes and has something totalitarian about it. It ignores the obvious question of

why some 97% of the population, many of whom have also been and continue to be exposed to adverse social influences, do not suffer from severe mental illness. The dogmatic denial of individual factors hinders further research, especially regarding the way in which social and economic factors significantly contribute to the individual illness that emerges and when and how those affected can be incorporated into therapeutic processes. The juxtaposition of societal misery and individual suffering and of individual and social psychiatry makes no analytical, political, pragmatic or moral sense. It misses the point of medicine as a healing art and craft. Viewing different discourses, paradigms and perspectives as contradictions often does not help. Individual medical assistance cannot wait until the commitment to prophylactic and structural change is successful.

### The skepticism towards science

Misguided philosophical and anthropological concepts are often rooted in misguided epistemology. Criticisms of the MGMH mix two weak epistemologies: naive realism and radical constructivism. Naïve realism refers to the physiological body, bodily medicine and social influences as such, while radical constructivism refers to cultural phenomena and, in a critical sense, repeatedly to the "psy disciplines" and the scientific gaze in general (Summerfield 2012; Mills et al. 2014; Mills 2015; Fernando 2018; Cosgrove et al. 2019). Science is seen as just another culturally conditioned perspective that is not allowed to make any particular claim to the truth of its statements (Mills, Fernando 2014).

In everyday life, we tend to get along with other patterns of interpretation. They have decisive advantages, especially in terms of their flexibility, their practical usefulness and their binding nature – something that is often guaranteed by their very indeterminacy. Scientific thinking and work have the advantage of critical engagement with hypotheses of all kinds, including one's own. Within its paradigms, science imposes conditions on what is to be considered true: It insists on logic, modeling, empirically and phenomenologically comprehensible data, parsimonious theorizing and the provisionality of all knowledge. That is a long list of conditions. It does not make sense nor is it justified to sideline other cultural patterns of thought and systems of interpretation and to declare them irrelevant. By the same token, there is no reason not to recognize the special effort of scientific research to make robust statements, whether in the sciences or the humanities.

### Respect and dialog

The negative view of the epistemic capacity of the other and of the hermenutic and scientific view is relatively new. For many years, psychoanalysis and ideology critique dominated the social sciences. They assumed that the subject is not master in its own house, that people and cultures do not need to know how they function, what their motives are, and much more. Hence, they saw the need for dialogical truth-telling (Buber, Gadamer, Habermas), from which an increasingly adequate knowledge of self and the world can emerge. The conversation itself is productive insofar as it is ideally open to the world and interested in truth. If one ascribes to the subject or the cultures complete competence in understanding themselves and their worldview, then the communication can only be a monological one. Both interlocutors may listen to each other, but they do not take the risk of a joint conversation that could question or at least relativize their view of things. This also means that they do not take themselves seriously as subjects who know

something and have something to say. They want to spare the other person at all costs, want to protect him from the wind of change, and do not trust him with the learning process. Respect becomes avoidance, the search for truth is surrounded by taboos, and the other is denied the recognition that the imposition of a serious confrontation would entail. This view of gaining knowledge is inevitably subjectivist or culturalist. The idea of epistemic as well as moral confrontation becomes superfluous because the truth is already slumbering in the subject's mind or within cultural views and only has to be awakened to speak. This view is more predominant the more distant the culture in question – and the easier it is to idealize it. Yet there is enough historical, ethnographic, and literary material for a critical view of one's own as well as of a foreign culture (Edgerton 1992; Gyaasi 2017; Graeber, Wengrow 2022).

It is an everyday occurrence for psychiatrists to listen to obviously unrealistic perceptions and interpretations, which then need to be questioned. At most, they will accept them temporarily, but otherwise, they will work towards strengthening the common view of the world and towards enabling the person concerned to lead an independent life within this common reality. A sufficient epistemological realism is very helpful for this and is philosophically justifiable.

From a dialogical point of view, the critical interlocutor can, from his or her other perspective, make a valuable contribution to the clarification and convergence of subjective and cultural worldviews, and dealing with mental health issues is no exception to the rule (Campbell and Burgess 2012; Gómez-Carrillo et al. 2020). A critical look at different perspectives may reveal that defence mechanisms, power and exploitation interests, anxiety and discrimination against minorities all play a major role.

Some views are false, whatever else they are. Even the appeal to divine inspiration is often a claim to dominion. There are no innocent cultures or subcultures. This is also true of the medical view. Multiperspectivity and dialogue seem fundamental to us. "The Movement [MGMH, M.H.] has a key role to play in calling for the recognition of multiple models in building a more inclusive global approach to mental health" (Campbell and Burgess 2012, p. 489).

### Psychiatry is in reality a system of power designed to control and eliminate deviant behavior

There are many reasons for viewing psychiatry in both theory and practice as a form of exercising power – ranging from rigidly applied diagnostics that lose sight of the individual patient through the "total institutions" (Goffman 1961) of the past as well as the present to the systematic extermination of mentally ill and disabled people under German National Socialism. Even in democratically organized societies, psychiatrists fulfil regulatory tasks in addition to therapeutic ones when they implement compulsory placements and carry out compulsory treatment. Much remains to be done despite all the progress made towards strengthening patients' rights, a partnership relationship between doctor and patient, and advanced facilities and treatments with fewer side effects. However, the critics of the MGMH again misjudge the current situation in psychiatry, especially in poorer countries, when they assume this dimension of psychiatry pertains to the whole. The existence of psychiatric hospitals in developing countries usually depends on the respective colonial history of the country. Thus, countries such as India, Pakistan, Sri Lanka or Guatemala have large psychiatric institutions with disastrous living and treatment conditions. The image of psychiatry as an apparatus of power is very much and

justifiably based on the experience of such institutions (Foucault, Goffman) and their dismantling has consumed and continues to consume enormous resources. There is also a great danger of psychiatric hospitals being built for visibility, for representativeness. Representative buildings express more in terms of power and prestige for some actors, regions and NGOs than do outpatient care and decentralized services. Where they exist and need to be funded, they consume a significant portion of the small mental health budget available while serving only a few patients (Magna and Yemoah 2018, WHO 2020).

Nevertheless, in many African countries such institutions play as little a role as does psychiatry as a whole. In Côte d'Ivoire, there are about 130 beds available for psychiatric patients, meaning one bed for every 200,000 inhabitants. There are currently about 30 practising psychiatrists in Côte d'Ivoire, but only five of them work outside Abidjan, the country's only city with over a million inhabitants. There are about 50 nurses trained in psychiatry (personal messages from Prof. Koua, head of the national psychiatry programme in Ivory Coast, 2022). In all West African countries, people with mental illnesses are very unlikely to come into contact with any psychiatric activity at all. By far the most important points of contact for patients are healers or Prayer Camps. We will come to those in a moment.

Critics of the MGMH claim that the use of scientifically oriented, ambitious psychiatric concepts is not only a misrepresentation of facts but also – in the spirit of neo-colonialism – suppresses and marginalizes indigenous interpretations and their ways of dealing with mental problems and problematic behavior. The local social psychiatric programmes can only work if patients, mental health professionals, family members, communities and society work together. It only makes sense if, above all, the patients individually and jointly represent their interests and are supported in doing so. However, if psychiatry is understood in biomedical terms only, it must follow a conception of man that is characterized by great passivity. Indeed, critics make it sound as if psychiatric practice consists only of coercive measures against patients frozen in passivity. "Target communities are not blank slates that sit waiting passively for external experts to come and solve their problems. They are active social agents, often exercising extraordinary courage and ingenuity in staying alive in adverse social circumstances, …" (Campbell and Burgess 2012, p. 388).

But every mental health worker knows that, in modern psychiatry, patients mostly have the final say and sooner or later determine their own treatment. In fact, the essential care problem in countries without general psychiatric care is to find those people in rural areas who need help in the first place. This is difficult. The problem is not to exert too much power, but to establish any contact at all with them, their relatives and the people who care for them, for example in Prayer Camps. This is indeed impossible without outreach work and the effort of translating and recognizing cultural differences and integrating them into psychiatric practice.

### A social psychiatric project in West Africa

How can we develop mental health care in LIC and LMIC that starts with a realistic picture of psychiatric practice? I would like to present a specific project in Côte d'Ivoire and, in particular, a specific aspect that clearly shows that meaningful psychiatric work in a West African country cannot be an import, it has to be West African. Let us take "West African" as a provisional construct. In Côte d'Ivoire, some 60 languages are spoken, religions coexist and

intermingle, kingdoms coexist with a reasonably well-functioning democracy, economic knowledge and scientific thinking are widespread, and people also participate in traditional rituals. Living conditions and cultures in African countries are both disparate and rapidly changing (Gureje and Ojagbemi 2019). But the term helps us draw attention to the position of this region in international psychiatric discourse, particularly the difference between Anglophone and Francophone African countries. Of the sixteen countries in West Africa, eight are Francophone. Four of them belong to the LIC and the other four to the LMIC group. Francophone countries operate even further below the radar of the global public, including the WHO and human rights-oriented NGOs and organizations. The language barrier seems to play an important role. Almost everything we have found in the research literature on GMH in West Africa refers to Ghana or Nigeria, sometimes the Gambia or Sierra Leone – all English-speaking countries. In Côte d'Ivoire, only one scientific article on mental health was published in 2019, compared with 34 papers in Ghana, which has half the population (WHO 2020b). Whatever is possible in a particular country also depends on how connectable it is to international discourse.

### The patients

The project I would like to report on briefly is called "Samentacom (Santé mentale communautaire/Community Mental Health)". It was initiated in Bouaké, in the centre of the country. Samentacom is mainly involved in helping people suffering from severe mental illness and epilepsy by community-based care. It is under the direction of Asseman Médard Koua, Professor of Psychiatry in Bouaké. When we speak of severe mental illness, we mean schizophrenia and other psychoses, bipolar and schizoaffective disorders, and severe depression. In the future, brain-organic disorders will certainly play a greater role (Spittel et al. 2019). Patients with epilepsy accounted for 18% of Samentacom patients in Côte d'Ivoire in 2021.

In Côte d'Ivoire, patients and their family members have to pay for treatment and medication. There are thousands of local health centres, called "dispensaries," in the country. Ivorian health centres are funded by the state and nurses are responsible for primary health care. However, they do not treat patients with severe mental illness or epilepsy. Samentacom staff try to motivate staff in the centres and work with them, demonstrating how to treat people with mental illness and epilepsy. When community-based mental health care is integrated with primary care it allows individualized and person-centred work within the familiar environment of families, neighbors, and health centres for many patients at lower cost (Patel et al. 2007, Mari et al. 2009, Jack et al. 2014, Barbui et al. 2020, Sugiura et al. 2020). Integration into primary healthcare allows also people who develop mental health symptoms due to treatable organic medical conditions such as infections to be identified and treated (Makin 2023). However, research evaluating decentralized community-based projects is difficult and not well-developed (Hanlon et al. 2010, Cohen et al. 2011).

In addition to local nurses, community health workers – so-called "agents de santé communautaires" – are of great importance to the project. They establish the link between patients and health centres by making home visits in the centre's catchment area. Since they are often on the road, they know the people in their area well and can establish and maintain the necessary contact. They are equipped with motorcycles and cell phones.

In 2021, a total of 3,714 patients with mental or epileptic illnesses received psychiatric and psychosocial treatment at the health centres supported by Samentacom.

But beneath all these considerations, a fundamental question arises: If the incidence for the target group of patients is – conservatively estimated – 4-5%, this means that there are at any time more than one million people in Côte d'Ivoire suffering from severe mental illness and epilepsy. Where are these people?

### Healers and prayer camps

The main points of contact for seriously mentally ill people and their relatives are traditional healers and so-called Prayer Camps. It is important to distinguish these two points of contact. Many people in Côte d'Ivoire trust traditional healers; they often have a good reputation and enjoy a recognized status in society. They are sought out for all kinds of problems, whether related to life in general, to health or to spiritual problems. We do not know how many healers are really trying to help people with serious mental illness. In 2016, a systematic review summarized 32 studies from 20 countries on the effectiveness of traditional healing methods in mental health. The authors concluded that "Some evidence suggests that traditional healers can provide an effective psychosocial intervention. Their interventions could help alleviate suffering and improve mild symptoms of common mental disorders such as depression and anxiety. However, there is little evidence that they alter the course of severe mental illnesses such as bipolar and psychotic disorders" (Nortje et al. 2016, p. 154). This is not to say that traditional healers cannot play an important role in the future of African psychiatry. Since traditional healers are considered trustworthy and their treatment is generally affordable for many people in West Africa, the obvious step is to develop cooperation between scientifically oriented psychiatry and traditional healers on the basis of mutual interest, understanding and support (Ae-Ngibise et al. 2010, Patel 2011; Musyimi et al. 2016; Gureje et al. 2015; Arias et al. 2016; Ojagbemi and Gureje 2020). "Promoting greater understanding, rather than maintaining indifferent distances may lead to more successful cooperation in future" (Ae-Ngibise et al. 2010, p. 558).

In contrast Prayer Camps (Camps de Prière) are village communities that also provide a spiritual centre for the population. They are usually led by commissioned or self-appointed spiritual authorities. They interpret mental illness as the result of a spiritual aberration and treat the sick with prayers, often by making them fast, thirst or vomit, sometimes with deliberate abuse in the hope that the evil spirits will leave the patients' bodies. Most reports on these conditions come from NGOs, investigative journalists and human rights organizations, not from scientific research (e.g. New York Times, 2015; Human Rights Watch 2020). Prayer Camps are certainly African, but you can see how such a designation is dazzling, because many align themselves with the evangelical organization CMA, a movement founded in the 19th century in the United States. The camps house varying numbers of mentally ill people, from a few to more than 100, many of whom are chained, tethered to trees outdoors, and can remain there for years. They often have no protection from animals, heat, downpours or people, and have no employment. The camps are lawless spaces, even if they violate national laws. Such Prayer Camps are also found outside of Africa. They have an asylum function in regions where there are no alternatives, similar to the old psychiatric institutions in Europe, but extremely dispersed and, unlike those huge institutions of the past, they have to be discovered first and recorded. The

outrage of many Ivorians is great when they learn what happens to mentally ill people in their country. During a recent visit by two representatives of our foundation, a senior health worker began to cry at the sight of pictures of people chained in the Prayer Camps.

### A pilot project for collaboration with the prayer camps

In 2020, we conducted a survey in Côte d'Ivoire, which was subsequently published in 2021 (Koua et al. 2021). The aim was to find out how many Prayer Camps actually exist in the defined area and what spiritual background they have. The survey started in the centrally located city of Bouaké and reached 541 Prayer Camps. Based on this number, we can extrapolate that there are approximately 2,000 Prayer Camps in Côte d'Ivoire. Of the 541 camps we found, 60.26% were evangelical Christian, 34.38% traditional, and ca. 5.36% Islamic (Koua AM, et al. 2021). It is important to keep in mind that in Côte d'Ivoire, a person can seek and practice spiritual guidance in different ways at the same time. None of the camp leaders had medical training. Forty-four percent had never been to school and 24% had attended only elementary school. Almost all of them worked in another profession to earn a living, mostly in agriculture. It was encouraging that more than half the camp leaders could envisage greater collaboration with outpatient mental health teams, whether in the form of medical and psychosocial information and training or specialized case management (see also Musyimi 2015). Very often, the leaders also wished for better equipment for their camps. We know from other studies and from our own experience that Prayer Camps tend to be favourably disposed towards psychiatric treatment for acute illnesses. Yet leaders often reject long-term treatment because it would interfere with the spiritual journey they believe is necessary (Arias et al. 2016). In any case, there should be an interest in regulating and registering some of these camps and closing others, monitoring the camps and preventing them from depriving mentally ill people of their freedom or from abusing them. But from the social psychiatric point of view, an attempt should be made to cooperate with as many Prayer Camps as possible.

In January 2023, we launched a 6-month project that established and evaluated close collaboration with 10 Prayer Camps in 2 regions of Côte d'Ivoire (Gbêkê and Nawa). The work took place primarily at the Prayer Camps, but also included follow-up with families and communities where possible. Necessary outreach activities was be implemented by a mobile outpatient team whose members are empathetic and familiar with local conditions. The project involved psychiatrists, psychiatric nurses and trained social workers who are primarily responsible for talking to patients and advising their relatives. There were intensive supervisions.

After 6 months, 35 of the 100 patients treated had left the camps. According to the reports of the staff and managers of the camps, these patients were in good health. Of the remaining 65, 55 were assessed as significantly improved. The number of chained patients fell from 21 in March 2023 to 0 in September 2023. We visited the project from March 23 to April 6, 2024 (Huppertz, Kroll 2024). I will describe a visit to one of the CdPs as an example: In a CdP near Soubré, 12 patients were admitted to the project. Seven have since been discharged. The responsible agent de santé and the head of the camp told us that the discharged patients had returned to their families in good condition. Five patients are still in the camp. We had a look at them and spoke to them. A young man had been chained up for 8 years and now had a mini-job in a store, but was only able to work to a limited extent. He complained of constant pain in his legs and

walked somewhat awkwardly, both presumably due to the long period of chaining. He had no significant psychopathology. Another young man had been chained for 4 years, was now very quiet, suffered from severe extrapyramidal symptoms, probably as a side effect of a depot injection 3 weeks ago. A small slender woman had been chained for 7 years, obviously previously confused. Now she was moving freely and seemed to be doing extremely well. A young man had been chained for 6 years, was now considered cured and had aspirations to become a nurse's aide. An older woman had never been chained, probably suffered from depression, but was psychopathologically unremarkable. In all four camps of the project that we visited, we found a similar situation or balance.

In our opinion, two patients had not improved. It must also be taken into account that a diagnosis is sometimes difficult without technical aid. In one patient, we had the impression of severe brain damage that had been present for 14 years. All negotiations and treatments, in which medication played a significant role, were carried out by local staff. When we visited the camps, we encountered moving gratitude from patients, relatives, project staff and even the so-called "prophets" (the spiritual leaders) of the camps.

They partly explained the good results by saying that their prayers got through to the patients much better thanks to the medication. Psychotherapists often have a similar view regarding the importance of medication for the effect of their interventions. It could also be important in a directly therapeutic sense to continue the cooperation with the prophets. There is no reason to limit psycho- and sociotherapeutic concepts globally to those concepts that are established in countries such as the USA or Germany. "If every system of psychotherapy depends on implicit models of personhood, which varies cross-culturally, then the goals and methods of therapeutic change must consider the cultural concept of the person." (Kpanake 2018, p. 198) If psychiatry is understood as a global network that must develop locally, then local healing practices and concepts must be incorporated into an emerging African psychiatry, as has been done time and again (Neki et al. 1986; Wilson 2013; Kpanake 2018; Wright and Jayawickrama 2021). I see no reason to stage this as a conflict with the MGMH unless one only trusts it with a "biomedical" approach (Wright and Jayawickrama 2021).

Of course, our experiences are not scientifically verifiable results, only casuistics. Our foundation has no money for more research. I can only repeat that more domestic and foreign research is urgently needed in these countries. In this way, we are responding to the call to give more space to qualitative, ethnographic research and also to be guided more in our questions by the realities in the field (Bayetti, Jain 2018, Koua 2022). However, it is equally important that therapeutic interventions are not set in stone, but are developed on the ground. We will also see how staff, both inside and outside the camps, handle the offer of drug treatment and counseling work with patients and their families. Will cooperation occur in practice? Will something new emerge in which better living conditions, prayers and narratives, diagnoses and prognoses, medications and conversations, spatial integration, employment and togetherness, rest and entertainment, the community in the Prayer Camps and the families complement each other, the internal and the external staff respect each other, set limits and cooperate? Will a common practice emerge? It can only emerge there, not in distant Europe, not in the academic institutions of African cities, and not in isolated non-public and lawless spaces. On the spot, starting with the material and immaterial conditions already given, traditional psychiatric practice – be it called "psychiatric" or not – can also

become a social art of healing that respects the people concerned, while calling on science to help.

## Conclusion: Situated psychiatry

In line with current philosophical and scientific theoretical concepts, I suggest psychiatry conceptualizing as a network of people, ideas, things, infrastructure and practices. In every complex practice and even in the emergence of every scientific finding, many actors contribute their practices, attitudes, ideas and ways of thinking. You will always try to make them compatible or even consistent and you will always fail. This applies in particular to interdisciplinary and intercultural collaboration. Psychiatry is exactly that, even if it takes place in a European country and even if you only think of subcultures within local populations. Also the Ivorian staff need translators in some areas just as much as we do when we treat refugees from Eritrea. The underlying reason for this is that in psychiatry, the natural sciences, humanities and social sciences intersect like perhaps in no other field.

Social psychiatry is inherently West African in West Africa; it cannot be called "Western." We should understand psychiatry not primarily in terms of theories and concepts, but in terms of interconnected practices. In such networks of people it is quite natural in practice and everyday life to look at a situation from different perspectives. This is first of all an enrichment that does not need to be expanded into a controversy or a polar view of either or as is currently common in the controversy between the MGMH and its critics (Campbell and Burgess 2012, Cooper 2016). Thinking in terms of similarities, blends, transitions and simultaneity enables us to overcome such simplistic juxtapositions and polarizations. Looking at situations, including their peripheries and backgrounds, reveals things that are all too self-evident and part of the conception and reality of "culturally informed mental health research and practice" (Chase et al. 2018).

It seems superficial and arrogant to think that "western psychiatry" could be exported to West Africa. African psychiatry will never be the result of a translation of biomedical research in the academic institutions of the developed world. It emerges from hellish roads and sandy tracks, the heat, Prayer Camps, traditional healers, village communities, many other diseases, pathic and active bodies, many languages, mopeds and their repairs, the publicity of the private, hospitality, joie de vivre, subsistence economy, lack of money as well as concepts and ideas. Scientific elements are but some of the many elements within these networks, no matter where in the world they are located. The influence of both the natural sciences and the humanities was – and is – very limited in the practice of psychiatry, even in developed countries. But when we learn that in some regions of the world people have to live in chains or tied to wooden blocks for years, then we should rally together – regardless of where we come from –, answer the humanitarian call, and apply a sceptical mind and creativity to doing something about it.

**Open peer review.** To view the open peer review materials for this article, please visit http://doi.org/10.1017/gmh.2024.120.

**Acknowledgements.** I would like to thank Wolfgang Krahl and Medard Koua, without whose inspiration and persistent work this text would not have been written.

**Author contribution.** Conception and Writing: M. H.

**Financial support.** No financial support.

**Competing statement.** The author is involved in the conceptualization and financing of the presented project in Côte d'Ivoire through the non-profit Mindful Change Foundation. There are no competing interests.

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
