## [Reviewer Report]

I was really intrigued to engage with the author’s ‘critique of the critiques’ here. I appreciate the author taking the criticisms that myself and others have made of the Movement for Global Mental Health Movement (MGHM) seriously and having some thoughtful reflections on them. It is of course incredibly important we have these debates over aspects of current mental health policy in and outside of the academy, so I applaud the author for being part of this process.

That said, I was incredibly disappointed at the level of discussion articulated here: the article is not well thought-through, it lacks systematic presentation, and there is frankly a lack of analytical substance to the argumentation (not helped by a poor use of evidence and lack of supporting academic scholarship). In the first half of the article, not only do the arguments of the critics come across as one-dimensional and dogmatic, the original philosophy and goals of the MGHM is actually treated in much the same manner. Because of this, the casual reader lacks a clear understanding of what the central tensions are on this issue. The argument given by the author then, unsurprisingly, appears to offer (a little) more nuance: it is proposed that psy-professionals are eclectic in practice and culturally aware, they use a variety of health models in their work, their treatment options go far beyond simply prescribing psychopharmaceuticals, they work together with local practitioners and care about these communities, and the work is humane and progressive in settings where people have been previously imprisoned and tortured for abnormal behaviour). To support this argument for the eclecticism of MGMH interventions on the ground, the author then presents a local mental health project in Côte d’Ivoire that they are part of. Yet the project is not being reported on here in evaluation form, but as what might take place once the service is operational (it began in January of this year). What seems to be reported as evidence here is actually a list of suggestions on what might happen. I found this another unconvincing part of the discussion which reflected the general drift over the course of the article from presenting evidence and having an academic debate on the MGMH and their critics to random assumptions and unreferenced statements on the fundamental caring nature of western psy-professionals and the mental health system in general.

In conclusion, this article does not do justice to the either the MGMH or their critics; much more systematic engagement with the central ideas, arguments, and evidence is required. Unfortunately I recommend this article be rejected. I have attached a copy of the article with more specific comments (and some additional references) which may be of help to the author in formulating a more robust piece of work on this topic in the future.

Specific comments:

pg 3 line 92-94: They obviously differ in specific orientation, but yes, broadly they are all power critiques of psychiatry. I take that point.

pg 3, line 96 - This is highly debatable; actually, a lot are psy-professionals. See highlight below.

pg 3 line 101-102: Summerfield, Fernando, Bracken and colleagues, and Cosgrove are all psy-professionals, usually psychiatrists.

pg 4, line 111: The Cohen ref could be more precise. You cite relevant chapters from the book separately which is good, but the whole book is not about MGMH. For my own critique see:

Cohen, B. M. Z. (2021) ‘A Postcolonial Critique of Mental Health: Empire and Psychiatric Expansionism’, in Moodley, R. and Lee, E. (eds) Routledge International Handbook of Race, Ethnicity, and Culture in Mental Health. Abingdon: Routledge, 32-42.

pg 4, line 120: Not true. Some critics do that for sure, but not all of us. The reason that we do particularly focus on the biomedical model, however, is because it dominates the current discourse (has since the 1980s). I know you disagree, but look at the vast numbers of psych-drugs being dispensed around the world right now...

pg 4 line 125-131: -Psychiatry cannot explain what ‘mental illness’ is, so this is all problematic, whether biological or socially-orientated. For the ongoing reliability and validity issues see, e.g.:

Allsopp, K., Read, J., Corcoran, R. & Kinderman, P. (2019) ‘Heterogeneity in Psychiatric Diagnostic Classification’, Psychiatry Research, 279: 15–22, https://doi.org/10.1016/j.psychres.2019.07.005

-What psychiatry claims to do and what it actually does are different things. Samson discusses the issues of ‘medico-eclecticism’ here:

Samson, C. (1995) ‘The Fracturing of Medical Dominance in British Psychiatry?’,

Sociology of Health and Illness, 17(2): 245–268.

pg 5 line 155-157: Kutchins, H., and Kirk, S. A. (1997) Making Us Crazy: DSM: Th e Psychiatric

Bible and the Creation of Mental Disorders. New York: Free Press.

pg 5, line 172-177: A number of issues: no references to support the argument here; justification for using labels of mental illness as being less problematic for socially deviant behaviour is back to front -type II error (see Scheff ref)- and is in no way adequate justification for medical/psych intervention anyway; this is also a rather cliched view of regional responses to socially deviant behaviour (behaviour which might actually be seen as quite positive locally depending on the cultural context).

pg 6, line 198-200: The biomedical model and the body/mind split still dominates psychiatry and the DSM. I applaud your belief that this is outdated, but its actually very much state of the art in academic psychiatry. I appreciate some psych people take different approaches, but this is very much at the margins of the profession...

pg 6, line 201-204: It is up to academic psychiatry to prove it - no one is denying people’s distress, but you can’t go forward as a serious medical profession without some evidence behind you. That’s the issue here - show us the convincing evidence for what psychiatry claims...

pg 6, line 212-213: This is an interesting statement because psych diagnostic categorizes have becoming more fluid over time (e.g ADHD symptomatology in each DSM), and are lacking such boundaries right now more than ever (see e.g Frances‘ ’Saving Normal‘ book, or Horwitz and Wakefield’s ’loss of sadness‘ book on the expanding borders of ’depression')

pg 6, line 218-219: We do this to avoid the reductionist language of ‘mental illness’ (as mentioned above, there’s a lack of validity to that construct)

pg 6, line 222-224: Just one ref from me, but many people have written on the subject, very pertinent to colonial psychiatry and beyond...

Cohen, B. M. Z. (2014b) ‘Passive-Aggressive: Māori Resistance and the

Continuance of Colonial Psychiatry in Aotearoa New Zealand’, Disability

and the Global South, 1(2): 319–339.

pg 7 238-241: No supporting reference here again. Is this is an argument for the biomedical model?

pg 7, line 247-249: In the case of vaccines, that argument works of course, with psych treatments it doesn’t (one has proved efficacy, one lacks it). (Of course psych drugs can sedate and that could be useful in itself!). Ignores the central argument we make - Psychiatry has been and is a political discipline, but shrugs it off as just doing medicine and helping individuals…

pg8, 284-287: Cohen, B. M. Z. (2016) Psychiatric Hegemony: A Marxist Theory of Mental Illness. London: Palgrave

Macmillan.

pg 9, 325-326: As there’s no scientific evidence, this becomes simply subjective and based on ideas of social deviance. Either this is a medical issue (science), or it is social deviance (culture), but both these positions cannot be argued simultaneously as I think is happening in this article.

pg 9, line 334-335: What psych as individuals think they’re doing for patients is different from what they actually do as a profession as a whole. I am really concerned about such statements - this doesn’t sound like an academic argument for MGMH anymore, but simply an appeal to the emotions. Please engage with the evidence and argumentation we and the MGMH use.

pg 9, line 338-339: Lane, C. (2007) Shyness: How Normal Behavior Became a Sickness. New Haven:

Yale University Press.

pg 10, line 348 - “The Movement” - MGMH?

pg 10, section 2.5 - Years for all refs required.

pg 11, line 386-397: Need more supporting refs again. ….And is this still about psych as power system?….

pg 12, line 429-430: Have you considered that these local treatments might ‘work’ for people compared to western treatments? Where’s the consideration of those studies?...

pg 12, line 442-443: Possibly it doesn’t need to be psychiatric at all? Its a shame we do not hear more about what such West African psychiatry looks like as opposed to western psychiatry...

pg 13, line 467: A little confusing - are you supporting the biomedical model here? I thought the approach being argued for was more eclectic than that.

pg 14, line 513-519: So is your point that we need GMH for SMI cases? Be good to state that clearly here.

pg 16, line 581-582: It would have been good to clarify that it is currently being theorized as to what the project might do here. What actually happens might turn out to be something quite different. Here’s an example:

Wright J & Jayawickrama J (2020) “We Need Other Human Beings in Order to be Human”: Examining the Indigenous Philosophy of Umunthu and Strengthening Mental Health Interventions. Culture, Medicine and Psychiatry https://doi.org/10.1007/s11013-020-09692-4

pg 16, line 591-595: Obviously all conjecture at this stage. I’m wondering how MGMH actually helps here and makes the difference - that needed to more central in this discussion.

pg 16, line 596-598: This needs to be outlined in the main discussion, rather than appear as new information in the Conclusion.

pg16, line 599-600: There’s no evidence of this here yet, but, yes, its a stated aim/hope for the project. Psychiatry is a western discipline, so why not consider the null hypothesis - that the intervention won’t improve things, and actually was not even necessary in the first place. I think its worth consideration given the issues with psych science I have mentioned above...

pg 17: There’s no evidence of this here yet, but, yes, its a stated aim/hope for the project. Psychiatry is a western discipline, so why not consider the null hypothesis - that the intervention won’t improve things, and actually was not even necessary in the first place. I think its worth consideration given the issues with psych science I have mentioned above...

---

## [Reviewer Report]

I see the first and second parts of this paper in different ways. The first part pertains to the underlying theories and practices that contribute to a polarizing discourse about the value and harms of the Global Mental Health Movement writ large. This part is somewhat incomplete and I do not think that it concisely or clearly conveys the issues. The second part, on the other hand, presents the rationale and practices of an attempt to create an outreach and treatment program in a community in Cote d’Ivoire. This second part is quite illuminating and an extremely useful example. It poses and partially answers some of the issues discussed in the first part, in a much more compelling way.

It is not unique in this respect, but certainly a potentially useful model for the region being discussed. Presenting this model in itself is a way of proposing a path through the polarizing discourse discussed in the first part. Therefore my suggestion would be to condense the first part to a 3-4 paragraph introduction, and retain the second half with virtually no changes required

---

## [Reviewer Report]

Respect for the people who are affected, a genuine interest in what they go through

and protection of their rights – these are the great concerns of traditional and modern

anti-psychiatry as well as social psychiatry: This may imply equating anti-psychiatry with social psychiatry. Please make the statement clear.

This list includes only the western psychiatrists. You may add global perspectives. The works of many like JS Neki, RL Kapur are valuable contributions from Asia.

3.1 The patients: Too many details. Please make this briefer.

---

## [Editor Report]

I have read through this document. The author took two perspectives - global perspective and Cote d’Ivoire perspective as a case study in Africa. It is gratifying to see a French speaking country used as an example in Africa and to see that what happens in this French speaking county is similar to what happens in most English speaking countries in Africa.

The write up flows. He has concluded that at the end of the day, all approaches to mental health must be context appropriate.

---

## [Reviewer Report]

My thanks to the author for spending time considering my comments and making the alterations to the article.

In the first round review I moved to reject on the basis that article did a lack of justice to both the MGMH or their critics, and that much more systematic engagement with the central ideas, arguments, and evidence (on both sides of the debate) was called for. In my opinion, the changes to the article which have since been made have added some nuance to understanding the debate around MGMH, and added some clarity to the author’s own position on this (for example, discussion on the similarities between antipsychiatry and social psychiatry, and the additions of Tables 1 and 2 regarding the central tenets of MGMH critics and medical model psychiatry). On the whole though, the article still requires major revisions. The reasons for this are:

• I am not convinced there is one coherent article on MGMH here. Either it is a piece considering (changing?) MGMH principles and their critics (for which there still needs to be more engagement with this critical literature on, for example, colonial and postcolonial psychiatry, capitalism, and the political economy of western mental health systems in the global south). Or it is a case study on the work of social/Westafrican psychiatry and the potential implications for MGMH. At the moment the article is unsatisfactory stretched between these two issues, with neither done justice (though I would welcome considered articles on both these topics).

• The evidence and supporting references for the author’s argument (and consideration of the alternative position) that psychiatry can still provide something of substance to the global south remains inadequate. In the last draft I suggested places where I thought references and more evidence was needed, some of this advice has been ignored (please compare the two versions of the article to see where such gaps remain). I am unconvinced that the author is fully engaging with the argumentation of critical scholars on MGMH, there is a reticent to consider research which disrupts the discourse of MGMH as an efficacious, humanitarian and progressive force, and I think that explains some of these omissions.

• Following the above, the article fails to consider the serious validity issues that critical scholars have raised with SMIs including schizophrenia, as well as the poor efficacy of treatments for SMIs including anti-psychotics, ECT, and psychotherapy (see for example the work of John Read). Even if western psychiatry achieved some efficacy here, there is also a lack of extended discussion on equating research on mental illness in the West with populations of the global south. Some discussion of the available research on what actually happens when MGMH lands in the global south (e.g. Wright and Jayawickrama, 2020) could have been very useful here as well. Consider also the null hypothesis again: can we compare experimental and control groups here, as well as long-term effects of an MGMH intervention? (I think we should, rather than just assuming it’s bound to be beneficial to someone’s ‘mental health’).

• Ultimately, I am surprised at a trained sociologist so easily ignoring the power dimensions of the psychiatric profession, the vested interests in the West of promoting MGMH in the global south, the economics of an expanding mental health system, and so on. A socio-historical glance at the mental health system (including colonial psychiatry) should raise serious concerns about the MGMH current claims to be humanitarian and progressive. We’ve seen this movie before, it doesn’t end well for people. For every person tied to tree, consider the iatrogenic deaths and injuries caused by psychiatry over time. One cannot be part of a logical argument for the other (though it has of course been used before within the colonial context; see for example Richard Keller’s work on Pinel and unchaining the mad in North Africa).

---

## [Reviewer Report]

Unfortunately I still find the first part disorganized and not very helpful as a publishable critique. The author makes points that I really appreciate and that are needed, but it is difficult to follow, is not concise, and does not convey a coherent overall perspective. The second part makes it easier to see the meaning, but the first part is still too long and unclear in my view. I wish the author could have done more to condense and clarify the points in the first half as it contains the seeds of many good ideas that could be developed

---

## [Editor Report]

This paper sets out to answer critics of the Global Mental Health Movement. These critics see Global Mental Health Movement as intrusive and taking over from other disciplines.

The authors of this paper argue that mental health is more inclusive than psychiatry as it sees individuals in their total social cultural environmental and economic contexts within the available and limited resources. Like any other medical condition, the determinants of mental health are multifaceted. The authors responded to all the criticisms by the 1st reviewer. In my view, the responses are polite and comprehensive.

This is a healthy debate. The authors have provided well-reasoned arguments to enrich the debate.